# Comparative Study of Quercetin and Hyperoside: Antimicrobial Potential towards Food Spoilage Bacteria, Mode of Action and Molecular Docking

**DOI:** 10.3390/foods12224051

**Published:** 2023-11-07

**Authors:** Mohamed Tagrida, Suriya Palamae, Jirakrit Saetang, Lukai Ma, Hui Hong, Soottawat Benjakul

**Affiliations:** 1International Center of Excellence in Seafood Science and Innovation, Faculty of Agro-Industry, Prince of Songkla University, Hat Yai 90110, Songkhla, Thailand; m.tagridaa@gmail.com (M.T.); suriya.pal@psu.ac.th (S.P.); jirakrit.s@psu.ac.th (J.S.); 2Key Laboratory of Green Processing and Intelligent Manufacturing of Lingnan Specialty Food of Ministry and Rural Affairs, College of Light Industry and Food, Zhongkai University of Agriculture and Engineering, Guangzhou 510225, China; m1991lk@163.com; 3Beijing Laboratory for Food Quality and Safety, College of Food Science and Nutritional Engineering, China Agricultural University, Beijing 100083, China; hhong@cau.edu.cn; 4Department of Food and Nutrition, Kyung Hee University, Seoul 02447, Republic of Korea

**Keywords:** antibacterial activity, hyperoside, quercetin, molecular docking

## Abstract

The antibacterial activities of quercetin and hyperoside were evaluated towards two major spoilage bacteria in fish, *Pseudomonas aeruginosa* (PA) and *Shewanella putrefaciens* (SP). Hyperoside showed a lower minimum inhibitory concentration (MIC) and minimum bactericidal concentration (MBC) towards both spoilage bacteria, PA and SP, than quercetin. Cell membrane morphology was affected when treated with hyperoside and quercetin. The release of content from the treated cells occurred, as ascertained by the release of potassium and magnesium ions and the increase in conductivity of the culture media. The morphology of cells was significantly changed, in which shrinkage and pores were obtained, when observed using SEM. Both compounds negatively affected the motility, both swimming and swarming, and the formation of extracellular polymeric substance (EPS), thus confirming antibiofilm activities. Agarose gel analysis revealed that both compounds could bind to or degrade the genomic DNA of both bacteria, thereby causing bacterial death. Molecular docking indicated that the compounds interacted with the minor groove of the DNA, favoring the adenine–thymine-rich regions. Thus, both quercetin and hyperoside could serve as potential antimicrobial agents to retard the spoilage of fish or perishable products.

## 1. Introduction

Fish are of significant nutritional and economical importance, thus making them an important diet component [1]. However, they are easily contaminated by spoilage microorganisms, especially under inappropriate storage and handling conditions. *Pseudomonas* spp. and *Shewanella* spp. are major spoilage bacteria in fish [2,3]. To prevent such a deterioration, different preservatives have been used [4]. Synthetic preservatives can extend the shelf-life of fish, but they have negative health impacts. This raises concerns for consumers who prioritize natural and organic food options [5].

Novel preservatives have been searched, particularly those derived from natural sources, as synthetic preservatives, which are commonly employed, not only pose significant health risks to consumers but also change the characteristics and sensory properties of the treated foods [6]. Plant polyphenols are diverse compounds with numerous bioactivities, e.g., anti-inflammatory, anticancer, antioxidant and antimicrobial activities [7]. Such compounds can be promising alternatives to synthetic preservatives. Additionally, green processing adopted by many organizations and governments worldwide can be fulfilled using natural preservatives [8].

Quercetin and hyperoside (quercetin 3-D-galactoside) are flavonoids found in many plants including tea, grapes, apples, onion, etc. [9]. Quercetin is a pigmented flavonoid that is synthesized from phenylalanine through the phenylpropanoid pathway using different enzymes [10]. Quercetin has shown antibacterial activities, as ascertained by its ability to suppress certain enzymes required by bacterial cells for their functions, causing bacterial death [11]. In one study, cork oak leaves (*Quercus suber* L.) containing a substantial amount of quercetin, along with other polyphenols such as epicatechin, gallic acid, catechin, rutin and myricetin, showed a substantial ability to control oxidation in cooked chicken. The treated samples had an extended shelf-life of an additional 5 days, compared to the control samples stored at 4 °C [12].

On the other hand, hyperoside is the glycoside derivative of quercetin, which is prevalent in several plants, e.g., hypericum, perforatum and barrenwort [13]. The structure of hyperoside is similar to that of quercetin except for the galactoside group, which connects to the main skeleton through an O-glycosidic bond. This unit can be cleaved through the action of β-galactosidase to liberate quercetin [14]. Hyperoside is a bioactive compound with bioactivities, e.g., antioxidant, antimicrobial, antitumor and anti-apoptosis activities, due to its active functional groups, particularly OH-groups [9]. Furthermore, hyperoside has exhibited antibiofilm properties against strains of *Pseudomonas aeruginosa* by means of suppressing the functions of some genes responsible for the formation of biofilm [15]. Apart from its antimicrobial potential, hyperoside has been used in some seafood products for quality improvement. Singh et al. [16] found that threadfin bream surimi gel with additions of hyperoside at 1% and 2% had the highest breaking force and deformation (*p* < 0.05). Water-holding capacity, hardness, springiness, gumminess and chewiness were improved after the addition of hyperoside to the surimi gel, signifying the enhancement of the textural and rheological properties of the surimi gel. In the same context, ethanolic extract of Duea ching fruit rich in hyperoside was effective in improving the gelling properties of sardine surimi gel [17]. The addition of this extract was found to lower the deterioration of the gel properties for 12 days of refrigerated storage, especially when packed under vacuum. 

These two phenolic compounds can be used as significant natural additives for the preservation of perishable foods to replace synthetic counterparts. However, the antimicrobial activities of hyperoside in comparison with quercetin have not been reported. Both compounds might have differences in antibacterial potential against common food-spoilage bacteria. For a better understanding of the bacterial inhibitory activity of both compounds, the mode of action and effectiveness of quercetin and hyperoside as preservatives against these spoilage bacteria must be elucidated. The aims of this study were to comparatively assess the antibacterial potential of quercetin and hyperoside toward spoilage bacteria and to investigate the mode of action of these two compounds using molecular docking. As a consequence, the potential phenolic compounds can be selectively used for preservation of perishable foods, especially seafoods and their products.

## 2. Materials and Methods

### 2.1. Materials

All chemicals (analytical grade) were purchased from Sigma-Aldrich (St. Louis, MO, USA). Culture media were bought from Oxoid (Thermo Fischer Scientific, Waltham, MA, USA). Quercetin (3,3′,4′,5,7-pentahydroxyflavone) with a purity of >95% was acquired from Yuanye Biotechnology Co. Ltd. (Shanghai, China). Hyperoside (quercetin 3- O -galactoside) was bought from Zelang Biological Technology Co., Ltd. (Nanjing, China). Both compounds were dissolved in 0.1% (*v*/*v*) dimethyl sulfoxide (DMSO) bought from Sigma-Aldrich (St. Louis, MO, USA). 

### 2.2. Study on Antibacterial Activities of Quercetin and Hyperoside on Spoilage Bacteria

#### 2.2.1. Bacterial Strains

*Pseudomonas aeruginosa* PSU.SCB.16S.12 was gifted from the Food Safety Laboratory, Prince of Songkhla University, Hat Yai, Thailand. *Shewanella putrefaciens* TBRC 5775 was procured from BIOTEC, Thailand. All the bacterial strains were kept at −80 °C in 50% glycerol. After being incubated (37 °C, 18 h) in tryptone soy broth containing 3% NaCl (*w*/*v*), the exponential phase with approximately 1 × 10^9^ CFU/mL was achieved.

#### 2.2.2. Minimum Inhibitory Concentration (MIC) and Minimum Bactericidal Concentration (MBC) of Quercetin and Hyperoside

The MIC and MBC of both compounds towards *Pseudomonas aeruginosa* (PA) and *Shewanella putrefaciens* (SP) were determined using the microbial dilution method [18]. The bacteria (100 µL, 1 × 10^6^ CFU/mL) were cultured in the presence of either quercetin or hyperoside (100 µL) at diverse concentrations (30–0.058 mg/mL) in sterile 96-well Microtiter™ microplates. Thereafter, the microplates were incubated at 37 °C for 24 h. Quercetin or hyperoside solutions without addition of bacterial suspension were used as negative controls, whereas the strains without the compounds acted as positive controls. Subsequently, 50 µL of resazurin solution (0.2 mM) was added to wells, and the MIC was identified as the lowest concentration yielding the complete inactivation of bacterial growth, where no color change was attained. Resazurin solution is a dark blue fluorogenic dye used as a redox indicator in cell viability tests. It undergoes a color change from dark blue (resazurin) to pink or purple (resorufin) when it is reduced, indicating the presence of metabolically active bacterial cells. The degree of color change is proportional to the number of viable cells and their metabolic activity [19]. Aliquots (15 µL) exhibiting no apparent bacterial growth were taken from the wells before the addition of the resazurin solution and inoculated into tryptic soy agar (TSA) plates. Thereafter, the TSA plates were incubated for 24 h at 37 °C. The lowest concentration that showed completely inhibited growth was considered as the MBC.

### 2.3. Study on Time–Kill Kinetics and Cell Leakage of Spoilage Bacteria for Quercetin and Hyperoside 

#### 2.3.1. Time–Kill Kinetics

Time–kill kinetics against both bacterial strains, PA and SP, were tested [1]. Quercetin or hyperoside solutions (1.5 mL) were inoculated with bacterial suspension (1.5 mL, 1 × 10^6^ CFU/mL), prepared as previously described in tryptic soy broth (TSB), to obtain the concentrations of 0.5, 1, 2 and 4 MIC. All samples were incubated (37 °C, 24 h) with continuous shaking. Surviving cells were enumerated after 0, 2, 4, 8, 12 and 24 h of incubation using the plate count method. Bacterial culture without any compound was used as the positive control. The detection limit was 10^2^ CFU/mL.

#### 2.3.2. Triphenyl-2H Tetrazolium Chloride (TTC) Dehydrogenase Activity

The TTC dehydrogenase activity of PA and SP treated without and with quercetin or hyperoside at 2 MIC was measured [20]. The TTC dehydrogenase relative activity (TRA) was calculated using the following equation:TRA (%)=AT/AU × 100
where *AT* and *AU* are the absorbance at 510 nm of the treated bacterial cells and the control, respectively.

#### 2.3.3. Potassium (K^+^) and Magnesium (Mg^2+^) Ion Leakage

The effect of quercetin or hyperoside at 2 MIC on the cell membranes of both PA and SP was evaluated by determining the released K^+^ and Mg^2+^ levels in the medium using inductively coupled plasma optical emission spectroscopy (ICP-OES) and reported as mg/L [21].

#### 2.3.4. Conductivity

The conductivity of the bacterial culture without and with treatment of quercetin or hyperoside at 2 MIC was determined [22]. The supernatants after centrifugation (3000× *g*, 10 min) of the bacterial culture were collected. The conductivity was determined using a conductivity meter (CON 200, CyberScan, Singapore City, Singapore).

#### 2.3.5. Malondialdehyde (MDA) Content

The thiobarbituric acid (TBA) method [23] was used to evaluate the lipid peroxidation taken place within the bacterial cells by determining the amount of malondialdehyde (MDA) generated. The supernatant (200 µL) obtained after centrifugation (7800× *g*, 10 min) of bacterial culture without and with quercetin or hyperoside treatment at 2 MIC was mixed with TBA solution (600 µL) and boiled for 10 min. After cooling down, the mixture was centrifuged (7800× *g*, 5 min), and the absorbance of the supernatant at 532 nm was read. For the calculation of MDA generated in the samples, a standard curve of MDA (0–10 nmol/mL) was employed.

#### 2.3.6. Scanning Electron Microscopic (SEM) Images

The morphology of untreated PA and SP and PA and SP treated with either quercetin or hyperoside at 2 MIC was visualized using a FEI Quanta scanning electron microscope (Hillsboro, OR, USA). A magnification of 20,000× was used.

### 2.4. Study on Anti-Swimming and Swarming Activity and Antibiofilm Formation of Quercetin and Hyperoside towards Spoilage Bacteria 

#### 2.4.1. Anti-Swimming and Swarming Motility

The abilities of PA and SP for swimming and swarming after the addition of quercetin or hyperoside were evaluated [22] using semi-solid swimming and solid swarming TSB plates, respectively. The diameter for the swimming test was determined after incubation for 2, 4, 6 and 8 h at 37 °C. The diameter was measured for the swarming test conducted after incubation for 6, 18 and 24 h.

#### 2.4.2. Biofilm Inhibition 

To determine the capability of quercetin and hyperoside to inhibit biofilm formation of PA and SP, the procedure of Mohamed et al. [24] was employed. Quercetin or hyperoside (100 µL) at different concentrations (2–1/4 MIC) was pipetted into the wells of 96-well flat-bottomed Microtiter™ microplates. Thereafter, 100 µL of bacterial culture (1 × 10^6^ CFU/mL) was dispensed into each well. Wells containing only TSB were considered as the negative control, while bacterial cell cultures without any compounds were regarded as the positive control. After incubation (37 °C, 24 h), the medium was carefully decanted, and the wells were washed two times using phosphate-buffered saline (PBS) pH 7.4, dried and stained with 0.1% (*w*/*v*) crystal violet solution for 20 min. The stain was drained, and the wells were washed with sterilized DW. Subsequently, each well was resuspended with 200 µL of absolute ethanol, and the absorbance at 570 nm was read.

#### 2.4.3. Biofilm Eradication 

Bacterial cultures (100 µL, 1 × 10^6^ CFU/mL) were inoculated into 96-well flat-bottomed microplates and incubated (37 °C, 24 h). Thereafter, planktonic cells were cautiously removed, and the wells were washed 3 times with PBS (pH 7.4). Next, 100 µL of quercetin or hyperoside at different concentrations (2–1/4 MIC) was added to the wells, which were then incubated (37 °C, 24 h). Subsequently, the washing, staining and absorbance measurement were performed as previously described. Negative and positive controls were treated in the same manner as that of the biofilm inhibition assay [24].

#### 2.4.4. Biofilm Cell Viability 

The viability of the biofilm cells was examined after 24 h of eradication treatment following the method of Mah [25]. The medium was first decanted, and the wells were rinsed with PBS 2 times. Subsequently, fresh TSB (100 µL) was added to all wells and incubated (37 °C, 24 h). Aliquots (5 µL) were taken from each well and inoculated onto the surface of TSA plates, in which the viability of the biofilm cells was measured using the growth of the inoculated aliquots.

#### 2.4.5. Extracellular Polymeric Substance (EPS) Content

The EPS content of bacterial cells was determined using the phenol sulfuric acid method [26]. Bacterial cultures treated with quercetin or hyperoside at 2 MIC were centrifuged (1100× *g*, 15 min). The supernatant (100 µL) was mixed with three volumes of ethanol (95%, *v*/*v*) and kept for 24 h at 4 °C for polysaccharide precipitation. After precipitation, 700 µL of phenol sulfuric acid reagent was added to the pellet and kept for 20 min at room temperature. Thereafter, the absorbance at 490 nm was read with the aid of a FLUOstar Omega microplate reader (Ortenberg, Germany). The reduction in the EPS content was calculated relative to that measured for the control (untreated) bacteria samples.

### 2.5. Study on Microbial DNA of Spoilage Bacteria Treated with Quercetin and Hyperoside 

#### 2.5.1. DNA Interaction

Genomic DNA was extracted from the selected bacterial cells using a DNeasy Blood & Tissue Kit (QIAGEN, Venlo, The Netherlands). The ratio of A_260_/A_280_ (1.8 ≤ A_260_/A_280_ ≤ 2.0) was calculated for purity [27]. Quercetin or hyperoside at varying concentrations (0, 1, 2 and 4 MIC) were mixed with the extracted DNA following the method of Palamae et al. [22]. DNA bands after treatment were visualized using a UVITEC documentation system (Fire-Reader XS, Cambridge, UK).

#### 2.5.2. Molecular Docking of Hyperoside–DNA and Quercetin–DNA Interactions

Computer simulation of molecular docking was used to elucidate the molecular interaction between hyperoside and DNA as well as between quercetin and DNA. A docking receptor (PDB ID: 453D) was downloaded from the Protein Data Bank (https://www.rcsb.org/structure/453D, accessed on 4 July 2023), and the molecular structure of hyperoside and quercetin were obtained from the ZINC database (http://zinc.docking.org/). Before docking, small ligand molecules and DNA molecules were water-deleted, hydrogenated and charge-added [28]. AutoDock Vina (Version 1.1.2) and AutoDockTools (Version 1.5.7) were used to simulate molecular docking between hyperoside and DNA and between quercetin and DNA. Specific docking was run for all ligands. A grid box centered at 21.712, −14.257, 10.574 with sizes of 30, 54 and 126 points along the XYZ directions with a spacing of 0.375 Å was generated to embrace the entire DNA molecule. The best DNA and ligand conformation were chosen from the lowest energy scoring of all docking results. The interactions and modes of binding between ligands and DNA were determined using Discovery studio, PyMol and LigPlot + V.2.1 software (European Bioinformatics Institute, Hinxton, UK).

### 2.6. Statistical Analysis

A completely randomized design (CRD) was employed, in which all experiments and analyses were performed in triplicate (*n* = 3). The results are reported as the mean ± standard deviation. One-way analysis of variance (ANOVA) was performed, and Duncan’s multiple range test (DMRT) was used to compare the means. Independent sample t-tests were adopted to compare two sample means. Data analysis was conducted using the SPSS package (SPSS 23.0 for Windows, SPSS Inc, Chicago, IL, USA).

## 3. Results and Discussion

### 3.1. Antibacterial Activity of Quercetin and Hyperoside toward Spoilage Bacteria

#### 3.1.1. MIC and MBC 

The antibacterial activity of hyperoside and quercetin is expressed as MIC and MBC against *P. aeruginosa* (PA) and *S. putrefaciens* (SP). Both compounds had similar antibacterial activity against PA and SP; MICs of 3.75 and 3.75 mg/mL and MBCs of 7.5 and 7.5 mg/mL were found, respectively. The antibacterial activity of both polyphenols is attributed to their functional groups, which can interact with bacterial cells and disrupt their structure and function [29]. Apart from the disruption of cell membranes, these compounds may exert their antibacterial activity via inhibiting bacterial enzymes or disrupting bacterial DNA [30]. In general, different responses towards the antibacterial compounds from varying bacteria may be governed by the contents of their cell walls and composition of their cells. MBC values (7.5 mg/mL) were observed for both compounds against PA and SP. Both bacteria are Gram-negative, in which their cell walls consist of peptidoglycan monomers, lipopolysaccharides, etc. [30]. Different compositions of cell walls could make Gram-negative bacteria susceptible to both compounds, leading to bacterial death [1].

#### 3.1.2. Time–Kill Kinetics 

The time–kill assay was used to study the antibacterial activity and determine the bactericidal or bacteriostatic activity of quercetin and hyperoside against SP and PA. At low concentrations (0.5 MIC), neither compound was effective in inhibiting the bacterial isolates, and their growth rates were not different from the control (Figure 1). Increasing the concentration of both compounds to 1 MIC, the growth rate of the bacteria was slightly diminished compared to the control. A reduction rate ≥ 3 log CFU/mL in bacterial count is regarded as bactericidal activity, while growth reduction less than this value is considered bacteriostatic [1]. Different classes of antibacterial agents affect the biosynthesis of peptidoglycans, making cells more susceptible to osmotic lysis [31]. Generally, antibacterial agents affecting the cell wall biosynthesis are bactericidal in their action [32]. Quercetin at 1 MIC had a reduction rate of 0.262 log CFU/mL against SP and 0.176 log CFU/mL against PA after 24 h of incubation, indicating bacteriostatic activity against both bacteria at this concentration. With augmenting concentrations, quercetin showed bactericidal activity after 4 h at 4 MIC and after 8 h at 2 MIC against SP. However, the bactericidal activity against PA took place after 12 h at 4 MIC and after 24 h at 2 MIC, in which there was no bacterial growth at these concentrations after the given periods of time. On the other hand, hyperoside was more effective in inhibiting bacterial growth. Both compounds have several OH groups; nonetheless, they are more prominent in the hyperoside structure due to the presence of the beta-D-galactosyl residue attached to its main skeleton [33]. At 1 MIC, hyperoside had a reduction rate of 1.138 log CFU/mL against SP and 0.381 log CFU/mL against PA after 24 h of incubation. The bactericidal activity at elevated concentrations of hyperoside was more pronounced than that of quercetin; the bacterial growth was totally inhibited after 2 h and 4 h at 4 MIC and 2 MIC, respectively, against SP. For PA, there was no noticeable growth after 2 h and 8 h at 4 MIC and 2 MIC, respectively. The different response of the bacteria against both compounds could likely be attributed to the different compositions of the bacterial cell wall. Both bacteria are Gram-negative, however the structure and composition of the lipopolysaccharide (LPS) layer in both bacteria could be different. As a result, these bacteria are resistant or have a barrier to certain groups of antibacterial agents at different degrees [34].

#### 3.1.3. TTC-Dehydrogenase Activity

The TTC-dehydrogenase activity of both bacteria decreased when treated with quercetin or hyperoside. The latter showed a greater reduction in enzyme activity than the former (*p* < 0.05). Viable bacterial cells possess dehydrogenases, which play crucial roles in cellular respiration, the synthesis of biomolecules and energy production [35]. These enzymes are able to convert TTC, a colorless low-molecular-weight compound, into triphenyl formazan (TF), a red hydrophobic compound that can be retained within bacterial cells [20]. Consequently, a decrease in TTC-dehydrogenase activity, indicated by a reduction in the formation of red TF, signifies a reduction in bacterial growth caused by the antibacterial agents. The results indicated that the viability of the bacteria decreased in the presence of both quercetin and hyperoside (Table 1). However, their effects and the responses of the treated bacteria differed significantly (*p* < 0.05). Hyperoside demonstrated superior effectiveness over quercetin in inhibiting the growth of *S. putrefaciens* or *P. aeruginosa* (*p* < 0.05). The presence of more OH groups in hyperoside might be the reason for its higher inhibitory activity, as evidenced by the lower TTC-dehydrogenase activity detected in treated cells. Both quercetin and hyperoside disrupted bacterial cell walls and negatively affected their permeability, thus lowering the balanced transportation of the contents between the cells and the environment. This disruption can lead to significant changes in the bacteria, ultimately resulting in their death [36].

#### 3.1.4. Leakage of Potassium (K^+^) and Magnesium (Mg^2+^) Ions

Treatment of both bacteria (SP and PA) with quercetin or hyperoside at 2 MIC led to the increased release of K^+^ and Mg^2+^ from the cells compared to the untreated cells (*p* < 0.05) (Table 1). This phenomenon was likely due to the distortions that took place in the cell membrane induced by the antibacterial compounds. Hyperoside was more effective than quercetin in causing the treated bacterial cells (SP and PA) to lose more K^+^ and Mg^2+^. This was likely owing to their different structures and functions toward the plasma membrane. Nevertheless, both compounds were able to alter the permeability of the bacterial cell membrane, causing higher leakage of more elements. Among the several constituents in the bacterial cell protected by the cell membrane, K^+^ and Mg^2+^ ions play important roles for the function of the bacterial cell. Potassium is a major intracellular cation in bacterial cells and its content is usually higher than other cations [37]. It is required for intercellular enzyme activities and helps in the maintenance of the membrane potential and the internal pH [38]. Magnesium, on the other hand, is used by bacterial cells for glycolysis, synthesis of peptidoglycans and replication of DNA [39]. The influx and efflux of these elements and other molecules through the bacterial cell are regulated by the plasma membrane. Leakage of these elements from the bacterial cell can occur if exposed to a compound with antibacterial properties of which the plasma membrane is the main target [20].

#### 3.1.5. Conductivity

The lowest conductivity value was observed in the untreated bacterial cells (*p* < 0.05) (Table 1). These values were amplified significantly when both bacteria were treated with either quercetin or hyperoside. The latter resulted in a higher value than the former, indicating higher potent effect toward both bacteria (*p* < 0.05). Such an increase in conductivity plausibly resulted from the increased release of intracellular ions related to the damage of the bacterial cell membrane when treated with quercetin or hyperoside. Conductivity determines the changes that take place in the cell membrane of the bacteria, particularly its permeability after exposure to an antibacterial agent. Conductivity can be used as an indication of the integrity of bacterial cells [25]. The results for the conductivity were in line with those of K^+^ and Mg^2+^ leakage. All the results confirmed the increased leakage of these ions along with other elements from the bacterial cells after exposure to both quercetin and hyperoside. The uncontrolled release of intracellular ions led to the increased conductivity of the cell culture associated with the released ions, such as K^+^ and Mg^2+^ [22,26].

#### 3.1.6. Malondialdehyde (MDA) Content

The MDA content increased considerably when both SP and PA were treated with either quercetin or hyperoside (*p* < 0.05) (Table 1). Bacterial cells treated with hyperoside had a higher content of MDA, compared to the control (*p* < 0.05). Malondialdehyde (MDA) is produced through the peroxidation of polyunsaturated fatty acids in cell membranes [40]. Usually, it has no functional importance to bacteria, but it can be used as a marker for oxidative stress and redox signaling and can serve as an indication for cell membrane damage [22]. This result signified that after the treatment with both quercetin and hyperoside, polyunsaturated fatty acids were released at a higher rate from the damaged cell membrane of SP and PA. The liberated fatty acids underwent oxidation, leading to a greater formation of MDA.

#### 3.1.7. Scanning Electron Microscopic (SEM) Images

The changes in the morphology of the bacterial cells (SP and PA) without and with treatment with quercetin and hyperoside were visualized using SEM. Untreated bacterial cells (Figure 2A,D) appeared intact and unharmed with a smooth surface, and no apparent damage was attained. The morphology of cells was changed drastically after treatment with quercetin, in that the bacterial cells appeared to be shrunken, deformed and full of perforation, as highlighted by white arrows (Figure 2B,E). These changes were more pronounced when the bacterial cells were treated with hyperoside, as the deformity and damage of the cells increased significantly and some bacterial cells appeared to have burst (Figure 2C,F). Such damage and deformities on the surface of the treated bacterial cells might explain the alterations in their permeability that led to the uncontrolled and increased leakage of the essential elements and micro-organelles. These phenomena eventually resulted in the retarded proliferation and inhibited growth of the bacterial cells. Similar behavior was observed in several bacterial cells treated with different antibacterial agents, particularly polyphenols [41,42]. Damaged cells coincided with the increase in intracellular ion leakage, conductivity and MDA content (Table 1).

### 3.2. Anti-Swimming, Anti-Swarming and Antibiofilm Activities of Quercetin and Hyperoside

#### 3.2.1. Anti-Swimming and Anti-Swarming Motility

Both SP and PA had broader swimming motility than swarming, which might be due to the more active movement of the polar flagella in contrast to the movement caused by the lateral flagella responsible for swarming movement. Additionally, the differences in key genes encoding chemotaxis proteins important for flagella gene expression as well as inactivity of the lateral flagella and lack of required biosurfactants have been reported to explain the difference in the motility behaviors of the two bacteria [43,44]. Treatment with quercetin or hyperoside affected the motility of both bacteria, as evidenced by the reduction in the diameter. The swimming (Figure 3A,B) and swarming (Figure 3C,D) motilities of SP and PA were significantly restrained when treated with both compounds, and this inhibition effect was in a concentration-dependent manner. Quercetin or hyperoside at 2 MIC had the highest impact on the swimming and swarming motility. After treatment of SP with the antibacterial agents, the swimming motility was reduced from 17.39% in the control to 5.71% at 2 MIC of hyperoside and to 8.69% at 2 MIC of quercetin after 8 h of incubation at 37 °C. Swarming motility was reduced from 72.09% to 36.24% at 2 MIC of hyperoside and to 47.82% at 2 MIC of quercetin for SP after 24 h of incubation at 37 °C. For *P. aeruginosa*, swimming motility dropped from 35.29% to 7.4% at 2 MIC of hyperoside and to 10.16% at 2 MIC of quercetin after 8 h of incubation at 37 °C. Swarming motility was lowered from 85.71% to 68.18% at 2 MIC of hyperoside and to 74.07% at 2 MIC quercetin. Higher effective ability of hyperoside to inhibit both types of movement in the bacteria was noted. The motility of pathogenic or spoilage bacteria is one of factors that lead to the colonization and biofilm formation on the host cells [22]. Usually, motility is carried out by the motility organelles on the bacteria, such as flagella, with either swimming or swarming movements known as the main prerequisite for biofilm formation [27]. Restraining the motility of bacteria might be a practical way to overcome their biofilm formation ability. The alteration in the motility behavior of the bacteria by quercetin or hyperoside is believed to be due to the disturbance in the electrochemical gradient of protons across the bacterial cell and within its organelles [45]. This results in changes in the synthesis of essential macromolecules used for the bacteria’s function, such as ATP synthesis, required for membrane transport and cell motility [46].

#### 3.2.2. Antibiofilm Activity

Both hyperoside and quercetin showed biofilm inhibition activity (Figure 4A) and the activity was in a dose-dependent manner. At 2 MIC, hyperoside inhibited the biofilm formation in SP and PA by 98.59% and 96.53%, respectively. The inhibition activity dropped to 25.41% and 22.74% in SP and PA, respectively, when ¼ MIC was used. Thus, both compounds at sub-inhibitory concentration were still able to reduce the formation of biofilm in the aforementioned bacteria. Quercetin at 2 MIC had a similar trend of inhibiting biofilm to hyperoside. Nevertheless, the inhibition activity was lesser than that of hyperoside, in that the biofilm inhibition was 92.01% and 89.12% in *S. putrefaciens* and *P. aeruginosa*, respectively. The main contributor to infections and antibiotic resistance as well as to the spoilage of many food materials is strongly connected with the formation of bacterial biofilm [47]. Bacterial biofilm functions as a protective layer for the bacterial communities in addition to its ability to adhere to different surfaces, thus promoting spoilage. To lower the ability of bacteria to form biofilm, an antibacterial agent not only inhibits the formation of the biofilm but also eradicates the biofilm [24]. The purpose of this biofilm inhibition strategy is to prevent biofilm formation, thereby reducing the number of viable cells that can attach to a surface and reform a biofilm [48]. On the other hand, biofilm eradication is required to disrupt the biofilm structure entirely and remove the embedded bacteria, thus reducing their viability within the biofilm matrix and preventing the biofilm from reformation [49].

Hyperoside at 2 MIC could eradicate biofilm by 85.04% and 82.36% in SP and PA, respectively, while quercetin at the same concentration had eradication activity of 78.4% and 75.44% against the corresponding bacteria (Figure 4B). The eradication activity of both compounds was observed to diminish significantly (*p* < 0.05) with decreasing concentration of antibacterial agents. Thus, the eradication activity of both compounds was in a dose-dependent manner. The eradication activity of both compounds was observed to be lower than their biofilm inhibition activity. The planktonic cells responsible for the formation of the biofilm matrix are the main targets of the antibacterial compounds. As a consequence, the biofilm formation can be inhibited in its early stages [47]. For the eradication of biofilms, a biofilm matrix already formed with extended growth periods contains complicated compounds, which are difficult to eliminate due to their high intrinsic resistance to antibacterial compounds [50]. Additionally, the EPS matrix acts as a barrier, preventing the antibacterial compounds from penetrating and affecting the biofilm cells. Thus, these biofilms can be maintained [24]. At 2 MIC of hyperoside, almost no bacterial growth for SP was observed, while approximately 64.35% of the viable cells of PA were eliminated (Figure 4C). Quercetin at 2 MIC was able to reduce the viability of SP and PA by 64.15% and 26.21%, respectively. This difference in cell viability can be related to the difference in the structure of both compounds, which affected the bacterial cells in different fashions. Moreover, quercetin might not be as effective as hyperoside in mitigating the biofilm EPS, leaving some cells unharmed. As a result, they could reform the biofilm matrix when the conditions were favorable for bacterial growth [51].

The results of antibiofilm formation were in tandem with those of motility (Figure 3). The motility of bacteria is one of the factors responsible for biofilm formation, as it enables the bacterial cells to move to certain surfaces and initiate the process of biofilm formation [26]. The other factor contributing to biofilm formation by bacterial cells is their EPS, which constitutes around 90% of the biofilm [47]. EPS acts as a network for transporting oxygen and other nutrients required to sustain the growth of the bacteria.

#### 3.2.3. Extracellular Polymeric Substance (EPS) Content

The bacterial production of EPS was affected when both bacteria were treated with quercetin or hyperoside (*p* < 0.05). A reduction in the EPS production of SP of 73% and 78% was observed after exposure to quercetin and hyperoside, respectively, while the production was reduced by 71% and 75% when using the same compounds on PA, respectively (Table 1). This denotes the effectiveness of hyperoside over quercetin in terms of inhibition of EPS production. Nevertheless, both compounds were able to affect the bacterial production of EPS, which can further lead to failure to form biofilms and inability to adhere to different surfaces, including the scales of the fish body surface [26]. Extracellular polymeric substances are the constitution material of bacterial biofilm, constituting 50% to 90% of the total organic matter of the biofilm, and either remain adhered to the cell outer surface or are released into its growth medium [52,53]. EPS acts as a physical barrier that interferes with the penetration of antibiotics or antibacterial agents through it and thus is considered as a critical and dynamic component of bacterial biofilm. This poses an obstacle in the eradication of biofilms that contain multicellular bacterial cells, in addition to supporting the bacterial adhesion to different surfaces [26]. In addition, EPS provides biofilms with immense mechanical plasticity, which serves as the protective shield that makes them able to grow in harsh conditions [54]. By effectively reducing EPS production, antibiofilm activity could be achieved. 

Based on different actions of both compounds, the biofilm formation can be reduced as proposed in Figure 5.

### 3.3. Microbial DNA Damages of Spoilage Bacteria by Quercetin and Hyperoside

#### 3.3.1. Interaction with Genomic DNA

The DNA band of the control (Figure 6A) was clearly observed. When hyperoside or quercetin at MIC and 2 MIC were used for treatment of both bacteria, the DNA band intensity was lower than that of the control. At 4 MIC, both compounds were able to degrade the genomic DNA of SP and PA, as ascertained by the absence of bands or the presence of bands with weaker band intensity than those treated with MIC and 2 MIC, particularly for PA. The DNA of PA was plausibly more resistant to the actions of the antibacterial compounds. Additionally, DNA bands with less intensity suggested that hyperoside or quercetin were not able to degrade the DNA completely, but only bound to it and caused alterations to it [27]. Flavonoids, including hyperoside and quercetin, can interact with DNA in various ways with the aid of their hydroxyl groups [55]. Besides attacking and altering the composition of the bacterial cell membrane, both hyperoside and quercetin might express their antibacterial behavior by binding to certain macromolecules within the bacterial cell, such as DNA [28]. DNA is regarded as one of the most essential molecules within the cell since it has responsibility for the synthesis of enzymes and other proteins essential for the physiological and metabolic processes of bacteria. The interaction of DNA with some polyphenols or derivatives might disturb gene expression and block the synthesis of different enzymes and receptors, thus depleting bacterial cell functions and furthering bacterial death [27]. When flavonoids bind to DNA, they form non-covalent interactions, and the charge of the flavonoid in the DNA-binding complex can be influenced by the specific interactions and the pH. The charge can vary from positive to negative depending on the nature of the interaction and the specific conditions [56]. This can bring about drastic impacts on the bacterial DNA, as shown in Figure 6A. This could lead to inhibition of bacterial viability [22].

#### 3.3.2. Molecular Docking

Molecular docking techniques are effective means for understanding, visualizing and predicting the nature and location of the interactions between DNA and other molecules [57]. Additionally, these techniques can estimate the energy used for molecule–DNA complexation [28]. Molecular docking of antibacterial compounds (hyperoside and quercetin) was carried out using a DNA double helix with the sequence of d(CGCGAATTCGCG)2 dodecamer (PDB ID: 453D) to predict the favored orientation and the binding capability of hyperoside and quercetin toward the DNA helix. The energetically most suitable docked conformation was chosen from 100 runs of docking simulations. As shown in Figure 6B,C, hyperoside was able to bind to the minor groove located within the base pairs of the DNA with a binding energy of −10.0 kcal/mol considered as the lowest-energy cluster. Quercetin could bind to a similar location on the DNA but with a binding energy of −10.7 kcal/mol, which was a lower-energy cluster between this compound and the DNA. Therefore, these docked conformations were selected for hyperoside–DNA and quercetin–DNA interactions, respectively. The minor groove of DNA is the target of many non-covalent binding agents. DNA binding with certain sequences, primarily A(adenine)- T(thymine), occurs through different forces, such as van der Waals interactions with the minor groove walls, hydrogen bonding to base pair edges and generalized electrostatic interactions [58]. The docking analysis revealed that hyperoside bound with the A–T-rich region of the DNA, mainly using hydrogen bonds for these bindings. Furthermore, the molecule could be surrounded by the base pairs of A14, A15, A19, T13 and T18, in which the favorable interaction between the functional groups of the DNA and the hyperoside took place [28]. Quercetin was also found to bind with the A-T-rich region. Nevertheless, this binding occurred using a combination of hydrogen bonding and van der Waals forces, in which quercetin was surrounded by the base pairs of A13, A14, T12 and T13, making these forces the favorable interactions between quercetin and the DNA molecule. The binding of hyperoside and quercetin to the A-T-rich regions of the DNA might be related to the narrower groove of these regions than that of G (guanine)- C (cytosine) regions, allowing the better interactions between the ligand and the groove walls of these regions [59]. Moreover, the aromatic rings in both compounds can facilitate torsional rotation to conform to the helical curvature of the minor groove, and subsequently, binding occurred by facilitating hydrogen bonding and van der Waals interactions [60]. Docking analysis revealed that both hyperoside and quercetin were able to bind to DNA through the groove, particularly to the minor groove of the DNA molecules using hydrogen bonds and van der Waals forces.

## 4. Conclusions

Hyperoside and quercetin had antibacterial activities towards spoilage bacteria, both SP and PA. Hyperoside generally showed higher activity (*p* < 0.05) than quercetin. Both compounds inhibited both SP and PA via different mechanisms, such as cell membrane disruption, permeability alteration and biofilm prevention. The compounds were also able to bind to DNA minor grooves via several bonds, especially hydrogen bonds and van der Waals forces. Therefore, both quercetin and hyperoside could be used to prevent the growth of spoilage bacteria, thus prolonging the shelf-life of perishable fish or other seafoods. Based on the findings of this study, both compounds showed remarkable antibacterial activities, thus having the promising tendency to be used as natural food preservatives, particularly for seafoods. Further investigation is required to apply the compounds in seafoods or other perishable products in which *Pseudomonas* spp. and *Shewanella* spp. majorly contribute to the spoilage. The shelf-life and consumer acceptability of products treated with these compounds must be evaluated for the full exploitation of these compounds as natural preservatives.

## Figures and Tables

**Figure 1 foods-12-04051-f001:**
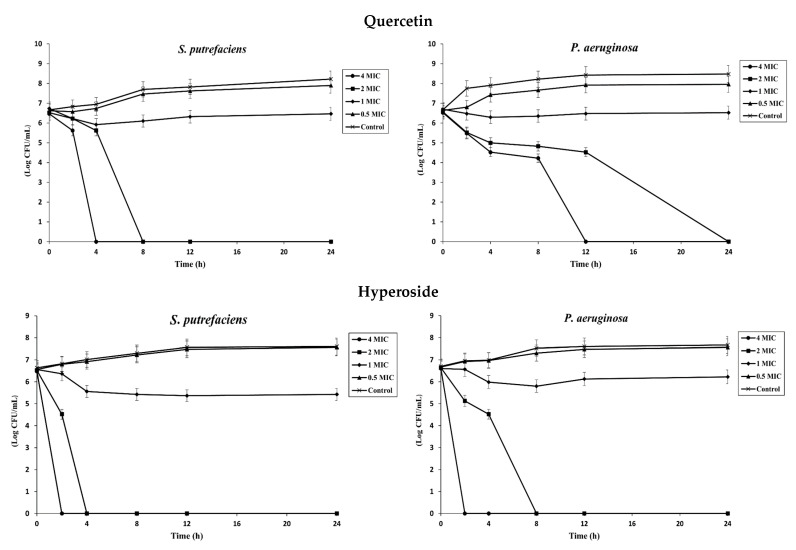
Time–kill profiles of quercetin and hyperoside at different concentrations against *S. putrefaciens* and *P. aeruginosa*. Bars represent standard deviation (*n* = 3).

**Figure 2 foods-12-04051-f002:**
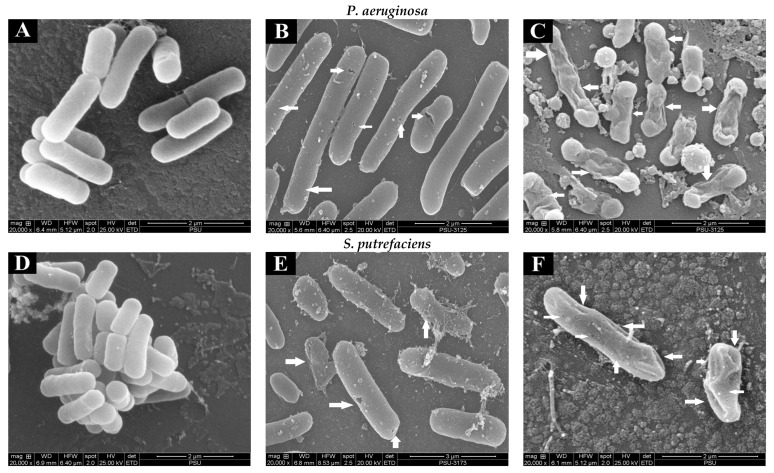
Scanning electron microscopic (SEM) images of *P. aeruginosa* and *S. putrefaciens* (untreated (**A**,**D**), treated with quercetin (**B**,**E**) and treated with hyperoside (**C**,**F**) at 2 MIC). Arrows indicate the damage of bacterial cells after being subjected to antibacterial agents.

**Figure 3 foods-12-04051-f003:**
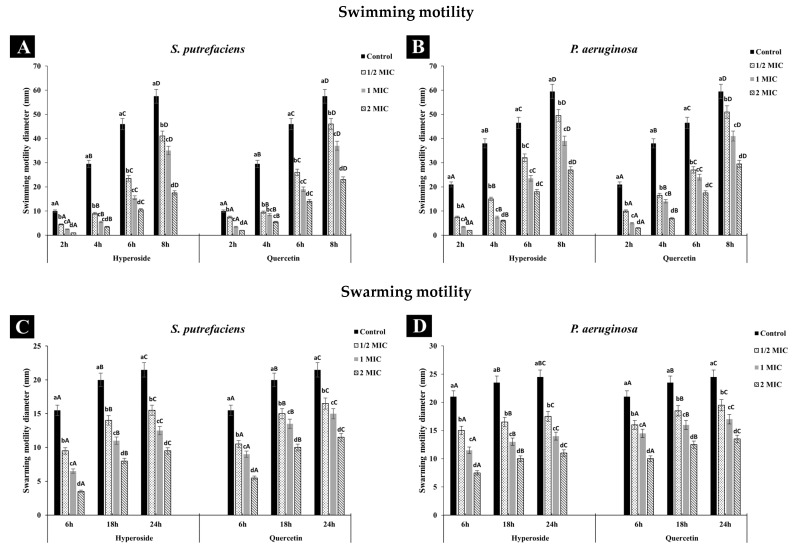
Effect of hyperoside and quercetin on the swimming and swarming motilities of *S. putrefaciens* (**A**,**C**) and *P. aeruginosa* (**B**,**D**). Different lowercase letters within the same time indicate significant difference (*p* < 0.05). Different uppercase letters within the same MIC of the same antibacterial agent indicate significant difference (*p* < 0.05). Bars represent standard deviation (*n* = 3).

**Figure 4 foods-12-04051-f004:**
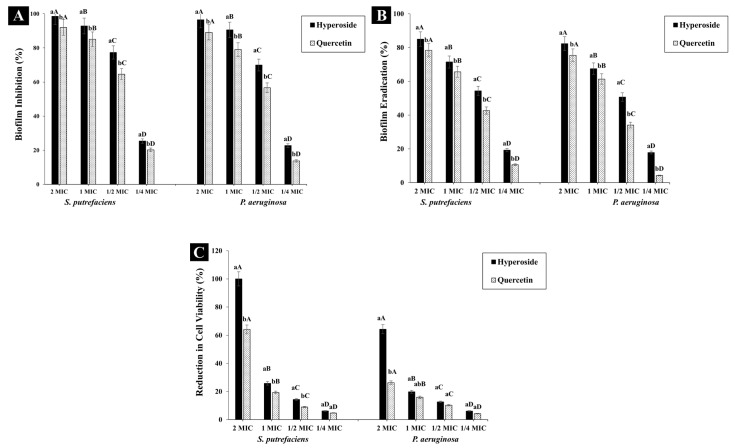
Biofilm inhibition (**A**), eradication (**B**) and cell viability percentage (**C**) of *S. putrefaciens* and *P. aeruginosa* after treatment with hyperoside and quercetin. Different lowercase letters within the same concentration indicate significant difference (*p* < 0.05). Different uppercase letters within the same antibacterial agent and target microorganism indicate significant difference (*p* < 0.05). The results are mean ± standard deviation (*n* = 3). Bars represent standard deviation (n = 3).

**Figure 5 foods-12-04051-f005:**
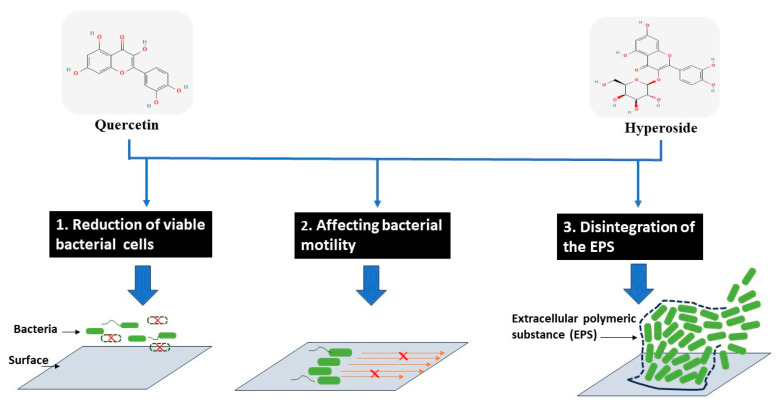
Modes of action for antibiofilm activity of hyperoside and quercetin against treated bacterial cells.

**Figure 6 foods-12-04051-f006:**
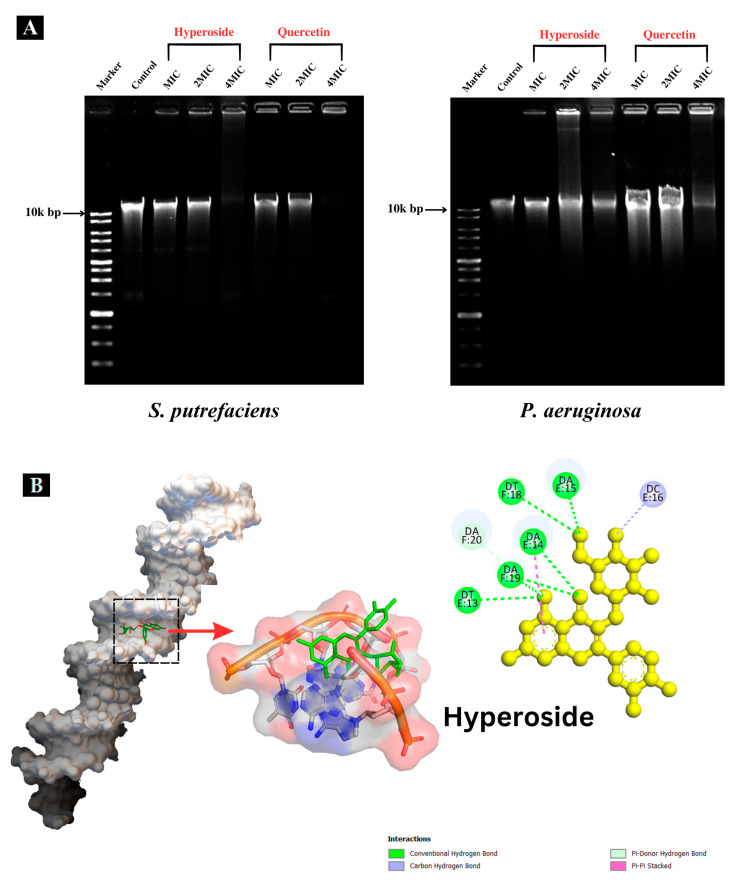
Interaction of hyperoside and quercetin with the genomic DNA of *S. putrefaciens* and *P. aeruginosa* analyzed using agarose gel electrophoresis (**A**). Molecular docking pattern of hyperoside (**B**) and quercetin (**C**) with DNA (PDB ID: 453D).

**Table 1 foods-12-04051-t001:** TTC-dehydrogenase activity, K^+^ and Mg^+2^ leakage, conductivity, MDA content and EPS content reduction of *S. putrefaciens* and *P. aeruginosa* (untreated and treated with quercetin or hyperoside at 2 MIC).

Activity	*Shewanella putrefaciens*			*Pseudomonas aeruginosa*		
	Control	Quercetin (2 MIC)	Hyperoside (2 MIC)	Control	Quercetin (2 MIC)	Hyperoside (2 MIC)
TTC- dehydrogenase activity (%)	88.22 ± 0.75 ^a^	37.31 ± 0.23 ^b^	30.64 ± 0.44 ^c^	90.36 ± 0.8 ^a^	52.74 ± 1.05 ^b^	41.43 ± 1.14 ^c^
K^+^ ion leakage (mg/L)	685.65 ± 0.63 ^c^	1182.4 ± 1.41 ^b^	1562.35 ± 1.48 ^a^	777.9 ± 0.56 ^c^	1208.65 ± 0.49 ^b^	1676.35 ± 1.34 ^a^
Mg^+2^ ion leakage (mg/L)	2.307 ± 0.01 ^c^	6.286 ± 0.01 ^b^	9.442 ± 0.07 ^a^	2.992 ± 0.01 ^c^	6.958 ± 0.06 ^b^	10.53 ± 0.01 ^a^
Conductivity (ms/cm)	4.75 ± 0.03 ^c^	8.11 ± 0.08 ^b^	12.55 ± 0.06 ^a^	4.97 ± 0.04 ^c^	9.02 ± 0.02 ^b^	13.7 ± 0.01 ^a^
MDA content (nmol/mL)	0.88 ± 0.01 ^c^	4.38 ± 0.02 ^b^	5.27 ± 0.01 ^a^	0.94 ± 0.01 ^c^	4.83 ± 0.01 ^b^	6.05 ± 0.01 ^a^
EPS content reduction (%)	-	73.04 ± 0.01 ^b^	78.07 ± 0.01 ^a^	-	71.55 ± 0.02 ^b^	75.18 ± 0.00 ^a^

Values are mean ± standard deviation (*n* = 3). Different lowercase letters in the same row within the same bacterial isolate indicate significant difference (*p* < 0.05). TTC: Triphenyl-2H tetrazolium chloride, MDA: malondialdehyde, EPS: extracellular polymeric substance.

## Data Availability

Data are contained within the article.

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
