# Peer review of "Comparative Study of Quercetin and Hyperoside: Antimicrobial Potential towards Food Spoilage Bacteria, Mode of Action and Molecular Docking"

_foods, 2023, doi:10.3390/foods12224051_

Round 1
Reviewer 1 Report
Comments and Suggestions for Authors
This manuscript discuss about Comparative Study of Quercetin and Hyperoside: Antimicrobial Potential towards Food Spoilage Bacteria, Mode of Action and Molecular Docking
An interesting knowledge has been reported. however the following comments should be addressed
comments.
The importance and significance of the work should be mentioned more clealry
Why authors choose Pseudomonas aeruginosa (PA), and Shewanella putrefaciens (SP).
Quercetin and hyperoside are previously well known for its biological activities. hence the novelty of the manuscript must be better emphasized
how does concentration of the compounds affects the biofilm and antimicrobial activity
for biofilm inhibition, the results of compounds dose not completely inhibited the development of biofilm. so it suggested to use some higher concentration to inhibit the biofilms
it suggested to add a mechanism based biofilm inhibition activity figure
update references, some of the references are old
enhance the quality of images
there are some typological errors are present in the manuscript that should be revised carefully
After addressing all the comments, this mansucript can be for further progress
Comments on the Quality of English Language
minor revision required
Author Response
Response to reviewer
Reviewer 1
*****Thank you very much for your insightful comments and suggestions. All queries have been responded and the required corrections have been made as highlighted in yellow.
The importance and significance of the work should be mentioned more clearly
*****The importance and significance of the work have been mentioned in the text. Please see lines 78 – 88.
Why authors choose Pseudomonas aeruginosa (PA), and Shewanella putrefaciens (SP).
*****The current work focused on evaluating the activity of the quercetin and hyperoside on the mentioned bacteria since they were the major spoilage bacteria in perishable foods, especially seafoods. Thus, authors decided to select those two major bacteria, mainly contributing to the seafood spoilage for current study.
The details on the role of two bacteria in the spoilage had been already provided in the introduction. Please see line 36-38.
Quercetin and hyperoside are previously well known for its biological activities. hence the novelty of the manuscript must be better emphasized
*****Some of the biological activity of quercetin and hyperoside had been already mentioned in the text (lines 50 – 68). The biological activities of both compounds, as mentioned, are adequately presented.
To our knowledge, no comparative study on the antimicrobial activities of both compounds against the important spoilage bacteria has been carried out. This finding provides new information and the significance of both compounds and their potential use as natural food preservatives. Thus, this manuscript has the novelty, providing new in-depth information including modes of action of both compounds in inhibiting the target spoilage bacteria to the readers.
how does concentration of the compounds affects the biofilm and antimicrobial activity
*****The concentration of antimicrobial compounds can significantly impact their antimicrobial activity including antibiofilm activity. Usually, higher concentrations of any antimicrobial agent tend to exhibit the increased effectiveness against the tested microorganisms. Due to a substantial dose of the active components, stronger inhibitory or bactericidal effects towards the treated microorganisms can be achieved. For example, the sufficient amount of active compounds could react with cell membrane, causing the leakage of content in the cells, etc. more effectively. However, it is worth noting that the optimal dose or concentration of the antimicrobial agent must be used to avoid the excessive amount of antimicrobial agent.
Hence, the minimum inhibitory concentration (MIC) and minimum bactericidal concentration (MBC) are required. Both MIC and MBC were examined in the current study against Pseudomonas aeruginosa and Shewanella putrefaciens. In addition, the relationship between concentration and antimicrobial activity can vary, depending on the antimicrobial compounds used, the type of target microorganism, and the intended application of the antimicrobial compounds, which was considered as food preservative in this study.
for biofilm inhibition, the results of compounds dose not completely inhibited the development of biofilm. so it suggested to use some higher concentration to inhibit the biofilms
*****As responded to the above query, the optimal dose or concentration of the antimicrobial compounds must be used, particularly in food preservation. Increasing concentration of antimicrobial compounds might not be necessary, since it plausibly can cause the undesirable changes in color or taste in treated samples, leading to consumer rejection.
In the present study, antibiofilm activity was conducted using in vitro study. In reality or real food system, the appropriate concentration must be optimized to inhibit biofilm due to the different surface characteristics. The insightful suggestion from the reviewer has been taken into consideration in our future work in the real system such as processing surface (e.g. cutting board, filleting area or sample surface, etc.)
it suggested to add a mechanism based biofilm inhibition activity figure
*****Thank you for your valuable suggestion. Actually, the exact mechanism of antibiofilm activity of quercetin and hyperoside is not clearly understood. However more studies to evaluate possible antibiofilm mechanisms of these compounds must be conducted in the upcoming works. Nevertheless, the possible mode of actions of both compounds against biofilm formation had been already introduced in the manuscript. Please see figure 4. Line 487.
update references, some of the references are old
*****The reference list has been checked and revised. The old references have been removed and the newer ones have been cited. Please see the list of references.
enhance the quality of images
*****The quality of the images has been improved as suggested. It is noted that the image quality may be compromised during the PDF file generation for peer review process.
there are some typological errors are present in the manuscript that should be revised carefully
*****Thank you very much for invaluable notes. The manuscript has been cross-checked carefully to remove any typological errors.
After addressing all the comments, this manuscript can be for further progress
*****Thank you so much for the invaluable comment, which can help the manuscript to be clearer and more informative.
Reviewer 2 Report
Comments and Suggestions for Authors
Dear Editor and Authors,
I send you my review about the article “Comparative study of quercetin and hyperoside: antimicrobial potential towards food Spoilage sacteria, mode of action and molecular docking”.
The scope of the paper, as reported in the aim was to assess antibacterial potential toward spoilage bacteria and mode of action using molecular docking of quercetin and hyperoside.
In my opinion, the paper is original, well structured and it is very interesting, however, it need of some little change that I report below.
The introduction is well written, but it is little concise respect the aim of the research. In my opinion, in this chapter it should report some research that have studied similar aspects of this research.
The chapter Materials and methods result well structured and complete. Nevertheless, to facilitate reading by readers it would be advisable to avoid excessive fragmentation in the subparagraphs. Therefore I suggest that the Authors limit themselves to the traditional division into chapters and paragraphs (such as 2.1, 2.2 etc.).
The results is very well presented and they are very well discussed, also in comparison to the data reported in the literature. However, also for this chapter I would like to suggest to Authors to merge the subparagraphs together.
Finally, the conclusions of the paper result adequate to the results showed and they satisfy the aim of the research.
Nevertheless, the section of the conclusions should not be limited only to reporting a summary of the data already reported, but should also include some personal comments of the Authors.
In this regard, I would suggest that the authors report their opinion on the impact that the results of their paper could have.
Best regards
Author Response
Response to reviewer
Reviewer 2
Dear Editor and Authors,
I send you my review about the article “Comparative study of quercetin and hyperoside: antimicrobial potential towards food Spoilage sacteria, mode of action and molecular docking”.
The scope of the paper, as reported in the aim was to assess antibacterial potential toward spoilage bacteria and mode of action using molecular docking of quercetin and hyperoside.
In my opinion, the paper is original, well structured and it is very interesting, however, it need of some little change that I report below.
*****Thank you very much for your encouraging opinions about our manuscript. All queries have been responded and the demanded amendments have been provided as highlighted in green.
The introduction is well written, but it is little concise respect the aim of the research. In my opinion, in this chapter it should report some research that have studied similar aspects of this research.
*****Thank you very much for your invaluable suggestion. Some studies with similar aspects to our work, especially the use of those compounds in foods, have been added. Please see lines 55 – 59 and 68 – 77.
The chapter Materials and methods result well structured and complete. Nevertheless, to facilitate reading by readers it would be advisable to avoid excessive fragmentation in the subparagraphs. Therefore I suggest that the Authors limit themselves to the traditional division into chapters and paragraphs (such as 2.1, 2.2 etc.).
*****Thank you very much for your suggestion. Although the fragmentation in the subparagraphs seems to be excessive, we assert that due to the substantial number of experiments, it is better to divide the methodology as sub-headings for better understanding. Actually, this approach is designed to enhance the readability and navigability of the study for the benefit of the readers. Thus, we prefer to keep the text in the present form.
The results is very well presented and they are very well discussed, also in comparison to the data reported in the literature. However, also for this chapter I would like to suggest to Authors to merge the subparagraphs together.
*****Thank you very much for your helpful suggestion. However, like the section of materials and methods, the fragmentation of this section into subparagraphs or subheading would improve the readability of the manuscript. Sorry, we would like to keep the ‘Result’ part in the present form.
Finally, the conclusions of the paper result adequate to the results showed and they satisfy the aim of the research.
Nevertheless, the section of the conclusions should not be limited only to reporting a summary of the data already reported, but should also include some personal comments of the Authors.
In this regard, I would suggest that the authors report their opinion on the impact that the results of their paper could have.
*****Thank you very much for your treasured suggestion. Opinions on the impact of the results of this study have been provided. Please see lines 586 – 592.
Reviewer 3 Report
Comments and Suggestions for Authors
This paper details the antibacterial activity of quercetin and hyperoside against P. aeruginosa and S. putrefaciens. Several interesting properties, including anti-biofilm activity, are described.
The following additions are requested.
1) Line 99: What is the resazurin solution used for? Please specify the reason.
2) Line 244-248: The statement here is misleading.
"The gram-positive bacteria were more resistant to antibacterial agents owing to the presence of a thick layer of peptidoglycan which prevent the penetration of the compounds through their cell walls."
Gram-negative bacteria that are more resistant to antibiotics than gram-positive bacteria are common; values for quercetin and hyperoside against gram-positive bacteria should be given and compared. Of course, other references can be cited.
3) Section 3.3.1 and 3.3.2: The authors show that quercetin and hyperoside may interact with DNA. If this mechanism of action is present in these agents, it could lead to the development of resistance and resistant strains. Please add whether other literature has shown resistance to quercetin and hyperoside.
Author Response
Response to reviewer
Reviewer 3
This paper details the antibacterial activity of quercetin and hyperoside against P. aeruginosa and S. putrefaciens. Several interesting properties, including anti-biofilm activity, are described.
*****Thank you very much for your beneficial comments on our manuscript. All queries have been responded and any demanded corrections have been provided as highlighted in turquoise.
The following additions are requested.
1) Line 99: What is the resazurin solution used for? Please specify the reason.
*****Resazurin solution is a dark blue fluorogenic dye used as a redox indicator in cell viability tests including minimum inhibitory concentration (MIC) test. It undergoes a color change from dark blue (resazurin) to pink or purple (resorufin) when it is reduced, indicating the presence of metabolically active bacterial cells. The degree of color change is proportional to the number of viable cells and their metabolic activity. For better understanding, the principle of resazurin used for testing has been provided in the text along with the reference. Please see lines 120 - 124.
2) Line 244-248: The statement here is misleading.
"The gram-positive bacteria were more resistant to antibacterial agents owing to the presence of a thick layer of peptidoglycan which prevent the penetration of the compounds through their cell walls."
Gram-negative bacteria that are more resistant to antibiotics than gram-positive bacteria are common; values for quercetin and hyperoside against gram-positive bacteria should be given and compared. Of course, other references can be cited.
*****We apologize for this unintended mistake. Initially the MIC and MBC tests were performed for different Gram-positive and Gram-negative bacteria in another study. However, we decided to focus more in the current study on the antibacterial behavior of quercetin and hyperoside against P. aeruginosa and S. putrefaciens since both bacteria are major food spoilage bacteria, and both are Gram negative bacteria. The discussion was written based on the initial assay that includes the characteristics of the Gram-positive bacteria. To remove unintentional mistake/misunderstanding statement, authors have removed it from discussion.
3) Section 3.3.1 and 3.3.2: The authors show that quercetin and hyperoside may interact with DNA. If this mechanism of action is present in these agents, it could lead to the development of resistance and resistant strains. Please add whether other literature has shown resistance to quercetin and hyperoside.
*****Thank you very much for raising this important point. To our knowledge, there are no reports about bacterial strains that developed certain resistance against either quercetin or hyperoside. Both compounds are of natural origin and their usage as antibacterial agents in various applications is relatively new. No information on the resistance against their actions has been documented. Therefore, no sufficient literature about any kind of resistance to these compounds are present.
In the contrary, both compounds were used to tackle the problem of the multi-drug resistance bacteria and the results of their antibacterial activity were reported by numerous literatures to show remarkable effects in inhibiting the growth of these microorganisms (Sun et al., 2017; Memariani et al., 2019; Nguyen & Bhattacharya, 2022; Zhang et al., 2022). Further investigations are needed to evaluate the prospected response of the continuous usage of quercetin and hyperoside for bacterial growth inhibition, to check if there will be any resistance of some bacteria toward these antibacterial agents.
References
Memariani, H., Memariani, M., & Ghasemian, A. (2019). An overview on anti-biofilm properties of quercetin against bacterial pathogens. World Journal of Microbiology and Biotechnology, 35, 1-16.
Nguyen, T. L. A., & Bhattacharya, D. (2022). Antimicrobial activity of quercetin: an approach to its mechanistic principle. Molecules, 27(8), 2494.
Sun, Y., Sun, F., Feng, W., Qiu, X., Liu, Y., Yang, B., . . . Xia, P. (2017). Hyperoside inhibits biofilm formation of Pseudomonas aeruginosa. Experimental and Therapeutic Medicine, 14(2), 1647-1652.
Zhang, Y., Liu, Y., Zhang, B., Gao, L., Jie, J., Deng, X., . . . Luo, J. (2022). A natural compound hyperoside targets Salmonella Typhimurium T3SS needle protein InvG. Food and Function, 13(19), 9761-9771.
Round 2
Reviewer 3 Report
Comments and Suggestions for Authors
The revisions were well done. This paper is acceptable.